# Retirement ages of senior UK doctors: national surveys of the medical graduates of 1974 and 1977

Fay Smith, Michael J Goldacre, Trevor W Lambert

## ABSTRACT

**Objective** To report on retirement ages of two cohorts of senior doctors in the latter stages of their careers.

**Design** Questionnaires sent in 2014 to all medical graduates of 1974 and 1977.

**Setting** UK.

**Participants** 3695 UK medical graduates.

**Main outcome measures** Retirement status by age at the time of the survey and age at retirement if retired. Planned retirement ages and retirement plans if not retired.

**Results** Of contactable doctors, 85% responded. 43.7% of all responding doctors had fully retired, 25.9% had 'retired and returned' for some medical work, 18.3% had not retired and were working full-time in medicine, 10.7% had not retired and were working part-time in medicine and 1.4% were either doing non-medical work or did not give details of their employment status. The average actual retirement age (including those who had retired but subsequently returned) was 59.6 years (men 59.9, women 58.9). Psychiatrists (58.3) and general practitioners (GPs) (59.5) retired at a slightly younger age than radiologists (60.4), surgeons (60.1) and hospital specialists (60.0). More GPs (54%) than surgeons (26%) or hospital medical specialists (34%) were fully retired, and there were substantial variations in retirement rates in other specialties. Sixty-three per cent of women GPs were fully retired.

**Conclusions** Gender and specialty differences in retirement ages were apparent and are worthy of qualitative study to establish underlying reasons in those specialties where earlier retirement is more common. There is a general societal expectation that people will retire at increasingly elderly ages; but the doctors in this national study retired relatively young.

## INTRODUCTION

In the UK, as in many other countries, as average life expectancy increases, retirement age and pensionable age will have to increase too. The World Economic Forum has reported that a baby born in 2017 can expect to live to over 100, and more immediately, retirement systems that were designed to support a retirement of 10–15 years will have to change to prepare for a 'seismic shift' as without substantially increasing

### Strengths and limitations of this study

► This is a large-scale study of graduates from all UK medical schools.
► The study has a very high response rate (85%) for a questionnaire study with voluntary participation.
► The study succeeded in differentiating between doctors who had fully retired and doctors who had taken formal retirement and returned in a medical capacity.
► However, there is a possibility of non-responder bias, in that retired doctors may have been less likely than doctors who had not retired to be easy to contact or inclined to respond.

retirement ages, individuals would eventually be spending more time retired than working.[1] Already changes are happening in the UK: for example, the state pension age is rising to 66 in 2020 and to 67 by 2028.[2]

Against this background and the trend to work longer to ensure financial security in old age, there are data which suggest that older doctors, both in the UK and in other countries, far from working longer, are in fact retiring early or reducing their working hours in increasing numbers in recent years.[3 4] A recent survey of consultant physicians in the UK found that their average intended age of retirement was 62 years and that 72% of these doctors did not intend to work beyond retirement age.[5] Early retirement of doctors has the major consequence for healthcare of early loss of doctors from the medical workforce.

In 2014, 20% of UK consultants and 21% of general practitioners (GPs) were aged 55 or over.[6] The median retirement age for GPs in the UK was 58 for women and 60 for men, between the years 2008 and 2011.[7]

The aim of this paper is to report on the retirement status and ages, and retirement intentions, of two cohorts of senior doctors who graduated in 1974 and 1977 whom we surveyed in 2014 when they had median ages of 64 and 61 years old, respectively. We

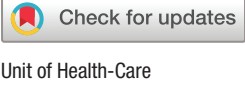

Unit of Health-Care Epidemiology, Nuffield Department of Population Health, University of Oxford, Oxford OX3 7LF, UK

**Correspondence to**
Dr Trevor W Lambert;
trevor.lambert@dph.ox.ac.uk

**Table 1** Questions and statements presented to respondents

| Asked of | Focus | Question/statement |
|---|---|---|
| Retirees and doctors who had 'retired and returned' | Retirement age | How old were you when you retired? |
| Doctors still working in medicine (full or part-time) | Retirement plans | At what age do you plan to retire? Do you plan to reduce your time commitments to your work before retiring? (Yes, I have already done so; Yes, in the future; No; Don't know) |

compared the replies of men and women, of those in the two cohorts and of those working in different career specialties.

## METHODS

The UK Medical Careers Research Group surveyed the UK medical graduates of 1974 and 1977. This was our last planned survey in career-long studies of the professional choices and work trajectories of these cohorts.[8–14] We sent postal and web-based questionnaires to these senior doctors in 2014 using current contact details provided to us by the General Medical Council. Up to four reminders were sent to non-respondents. Further details of the methodology are available elsewhere.[15]

The surveys sent to both cohorts were identical and comprised structured, 'closed' questions and statements. Doctors were asked to indicate their current employment status, choosing from seven categories (working in medicine full-time or part-time, working outside medicine full-time or part-time, retired, 'retired and returned' for some medical work or other). Doctors were then asked to respond to questions about their retirement age (if they had retired) or their retirement plans (if still working; see table 1).

We allocated a career specialty to each respondent using their recorded job history as reported to us in successive surveys and additional information about their specialist registration with the Genereal Medical Council as reported to us by the doctors in these surveys. For a small number of respondents, we were unable to allocate a single-career specialty, either because we did not have sufficient data about the doctor's career or because the doctor had worked in different specialties during their career. The career specialty allocation allowed us to analyse the responses and employment status of doctors in different specialties. Respondents were then grouped for analysis into seven groups: general practice, hospital medicine (comprising the hospital physician specialties and paediatrics), surgery (comprising the surgical specialties, emergency medicine, obstetrics and gynaecology and clinical oncology), anaesthesia, psychiatry, pathology and radiology.

The replies were analysed using $\chi^2$ tests and Mann-Whitney U tests to explore differences between male and female doctors, between cohorts and between doctors working in different specialties.

During ethical approval, we obtained permission to approach doctors for participation and to take their response as an indicator of informed consent to participate in the studies.

### Patient and public involvement

Patients were not involved in the design or any aspect of the study, by agreement with the funding body, since the study did not involve any medical or patient data. Results of the study are published in various papers in the peer-reviewed literature, and summary reports of our survey work are on our website at www.ndph.ox.ac.uk.

## RESULTS

### Response rates

We were able to contact 4369 of the original 5482 UK medical graduates from 1974 and 1977 (table 2). The aggregated response rate from both cohorts was 84.6% (3695/4369). To maximise response, at the end of the survey period, non-responders were given the option of completing a shortened questionnaire: 98 did so, and because of the curtailed information obtained from them, they could not be included in some of the analysis below.

### Employment status in 2014

In 2014, 43.7% of all responding doctors had retired, 25.9% had 'retired and returned' for some medical work, 18.3% were working full-time in medicine, 10.7% were working part-time in medicine and 1.3% were either doing non-medical work or did not give details of their employment status (table 2). Having graduated 3 years earlier than the 1977 graduates, more doctors from the 1974 cohort (52.0%) had retired than from the 1977 cohort (37.7%).

In further analysis, we excluded doctors whose career specialty (see Methods section) was unknown, or who were working outside medicine, or who did not declare their employment status, and analysed the employment status in 2014 of the 3435 respondents remaining, by gender and specialty grouping (table 3).

Taking men and women together, employment status and career specialty were strongly related ($\chi^2_{18}$=324.1, p<0.001): the most notable specialty differences were that more GPs (54%) and fewer surgeons (26%) and hospital medical specialists (34%) were fully retired compared with the overall average, and fewer GPs (10%) and more

**Table 2** UK doctors who graduated in 1974 and 1977: response to survey and career status of respondents

| Target grouping | Year of graduation | | |
| --- | --- | --- | --- |
| | 1974 | 1977 | Total |
| Graduation cohort | 2347 (100%) | 3135 (100%) | 5482 (100%) |
| Known to be deceased | 100 (4.3%) | 110 (3.5%) | 210 (3.8%) |
| Declined to participate | 20 (0.9%) | 50 (1.6%) | 70 (1.3%) |
| Uncontactable | 415 (17.7%) | 418 (13.3%) | 833 (15.2%) |
| Contacted | 1812 (77.2%) | 2557 (81.6%) | 4369 (79.7%) |
| Contactable doctors | 1812 (100%) | 2557 (100%) | 4369 (100%) |
| Did not respond | 267 (14.7%) | 407 (15.9%) | 674 (15.4%) |
| Responded in brief | 47 (2.6%) | 51 (2.0%) | 98 (2.2%) |
| Responded in full | 1498 (82.7%) | 2099 (82.1%) | 3597 (82.3%) |
| Respondents | 1545 (100%) | 2150 (100%) | 3695 (100%) |
| Working full-time in medicine | 195 (12.6%) | 483 (22.5%) | 678 (18.3%) |
| Working part-time in medicine | 145 (9.4%) | 252 (11.7%) | 397 (10.7%) |
| Working full-time outside medicine | 6 (0.4%) | 10 (0.5%) | 16 (0.4%) |
| Working part-time outside medicine | 7 (0.5%) | 9 (0.4%) | 16 (0.4%) |
| Retired not now working in medicine | 804 (52.0%) | 810 (37.7%) | 1614 (43.7%) |
| Retired and 'returned' for some medical work | 380 (24.6%) | 577 (26.8%) | 957 (25.9%) |
| Other/no reply | 8 (0.5%) | 9 (0.4%) | 17 (0.5%) |

surgeons (33%) and hospital medical specialists (29%) were working full-time compared with the overall average (based on analysis of adjusted residuals).

Comparing specialties separately for men and women (table 3), the specialty differences in employment status for men were significant ($\chi^2_{18}$=2241.7, p<0.001) and in the same specialties as the total. For women, the specialty differences in employment status were also significant ($\chi^2_{18}$=84.3, p<0.001); but the percentage of women hospital medical specialists who were retired (54%) was much closer to the percentage of women GPs (63%) who were retired than was the case for the men (24% vs 50%).

Online Supplementary tables 1 and 2 show results for the 1974 and 1977 cohorts separately. In many cases, the numbers in individual specialties are too small for meaningful statistical comparisons to be made.

### Age at retirement
Figure 1 shows the percentages of men and women respondents in each cohort who had not retired and who were working in medicine at each year of age. (The figure treats 'retired and returned' as not working in medicine for this purpose.) Log-rank tests showed that the male–female difference was significant in the 1974 cohort ($\chi^2_1$=93.9, p<0.001) and in the 1977 cohort ($\chi^2_1$=78.8, p<0.001): at any age, the percentage of women who were retired was larger than that of men. Within gender, the two cohorts followed similar patterns of retirement by age (women $\chi^2_1$=0.3, p=0.57; men $\chi^2_1$=0.7, p=0.39).

Figures 2 and 3 show the data for men and women separately, subdivided by career specialty, based on the seven groups previously defined (see Methods section). The cohorts have been combined but the gender separation has been retained, in view of the results from figure 1. Specialty differences were more marked for men ($\chi^2_1$=130.7, figure 2) than for women ($\chi^2_1$=25.1, figure 3), both p<0.001. Within each specialty, the gender difference was significant with p<0.001 for general practice and hospital medicine and was borderline significant for surgery, anaesthesia and pathology (in each case with higher percentages of women than men having retired): there was no gender difference for psychiatry or radiology (both p>0.05).

The average actual retirement age (including those who had retired but subsequently returned) was 59.6 years (59.9 men, 58.9 women; 60.4 1974 cohort, 58.9 1977 cohort). Psychiatrists (58.3) and GPs (59.5) retired at a slightly younger age than radiologists (60.4), surgeons (60.1) and hospital specialists (60.0) ($\chi^2_6$=67.8, p<0.001; Kruskal-Wallis test).

### Doctors who had not retired and who were still working in medicine: planned retirement ages
The average planned retirement age of doctors still working was 65.7 years (66.0 men, 64.7 women; 67.4 1974 cohort, 64.9 1977 cohort; 65.9 full-time, 65.3 part-time). Psychiatrists still working planned to retire at an older age (67.4) than doctors in other specialties: anaesthetists (64.5), pathologists (64.9), GPs (65.2) and radiologists (65.2) ($\chi^2_6$=26.0, p<0.001; Kruskal-Wallis test).

### Doctors who had not retired and who were still working: retirement plans
When asked 'Do you plan to reduce your time commitments to your work before retiring?' (38.2%) of doctors who were still working answered 'Yes, I have already

**Table 3** Career specialty and employment status at the time of survey of graduates of 1974 and 1977

| | Employment status | | | | |
|---|---|---|---|---|---|
| | Full-time in medicine | Part-time in medicine | 'Retired and returned' | Retired | Total N (100%) |
| | % | % | % | % | % |
| **Total** | | | | | |
| | 18.4 | 10.6 | 26.7 | 44.3 | 3435 |
| General practice | 10.1 | 13.6 | 22.0 | 54.2 | 1676 |
| Hospital medicine | 28.7 | 9.3 | 28.5 | 33.5 | 582 |
| Surgery | 33.3 | 7.5 | 32.7 | 26.4 | 504 |
| Anaesthesia | 19.2 | 6.1 | 27.1 | 47.7 | 214 |
| Psychiatry | 15.1 | 7.3 | 40.6 | 37.0 | 192 |
| Pathology | 17.7 | 5.7 | 28.5 | 48.1 | 158 |
| Radiology | 25.7 | 7.3 | 33.0 | 33.9 | 109 |
| **Women** | | | | | |
| | 10.5 | 12.0 | 20.6 | 56.9 | 1033 |
| General practice | 6.0 | 14.4 | 16.1 | 63.4 | 547 |
| Hospital medicine | 15.8 | 8.9 | 21.6 | 53.7 | 190 |
| Surgery | 23.6 | 11.1 | 26.4 | 38.9 | 72 |
| Anaesthesia | 17.3 | 3.8 | 17.3 | 61.5 | 52 |
| Psychiatry | 10.0 | 10.0 | 42.2 | 37.8 | 90 |
| Pathology | 7.5 | 9.4 | 20.8 | 62.3 | 53 |
| Radiology | 20.7 | 13.8 | 24.1 | 41.4 | 29 |
| **Men** | | | | | |
| | 21.8 | 10.0 | 29.3 | 38.9 | 2402 |
| General practice | 12.1 | 13.2 | 24.9 | 49.8 | 1129 |
| Hospital medicine | 34.9 | 9.4 | 31.9 | 23.7 | 392 |
| Surgery | 35.0 | 6.9 | 33.8 | 24.3 | 432 |
| Anaesthesia | 19.8 | 6.8 | 30.2 | 43.2 | 162 |
| Psychiatry | 19.6 | 4.9 | 39.2 | 36.3 | 102 |
| Pathology | 22.9 | 3.8 | 32.4 | 41.0 | 105 |
| Radiology | 27.5 | 5.0 | 36.3 | 31.3 | 80 |

Percentages are row percentages. Excludes 32 respondents working in non-medical jobs and 228 whose employment status and/or career specialty was unknown.

See Method section for explanation of career specialty grouping.

For detailed specialty breakdown in each cohort, with numbers and percentages, see Supplementary tables 1 and 2.

done so', 29.2% answered 'Yes, in the future', 22.2% answered 'No', and 10.4% answered 'Don't know'. More men (32.2%) than women (19.5%) answered 'Yes, in the future' ($\chi^2_4$=21.5, p<0.001). There were significant differences between specialties ($\chi^2_{18}$=69.9, p<0.001). More GPs (53.1%) answered 'Yes, I have already done so' than surgeons (29.7%) and doctors in hospital medicine (29.3%).

## DISCUSSION
### Main findings
Given that these doctors were predominantly only in their early 60s, the level of retirement was surprisingly high given societal changes in life expectancy and retirement planning. 'Retiring and returning', often to a less demanding or less time-committed post, and substantial levels of part-time working among the remainder, meant that there remained only a small minority of these highly experienced doctors working full-time in medicine.

Specialty differences in retirement levels at the time of the survey were seen for both men and women. They were more pronounced for men, with male hospital doctors having low levels of retirement compared with other men or women. This may reflect gender roles and expectation in this older generation, factors which may not apply to

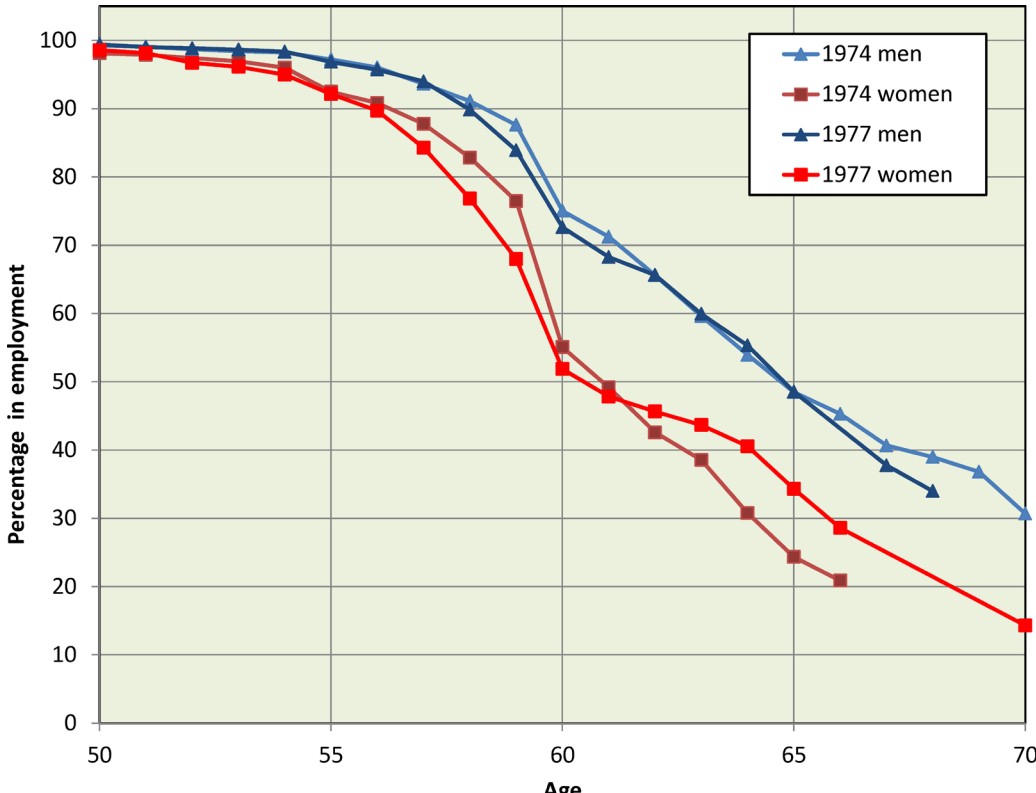

**Figure 1** Percentage of 1974 and 1977 graduates who had not retired and who were working in medicine, by age: Kaplan-Meier estimates

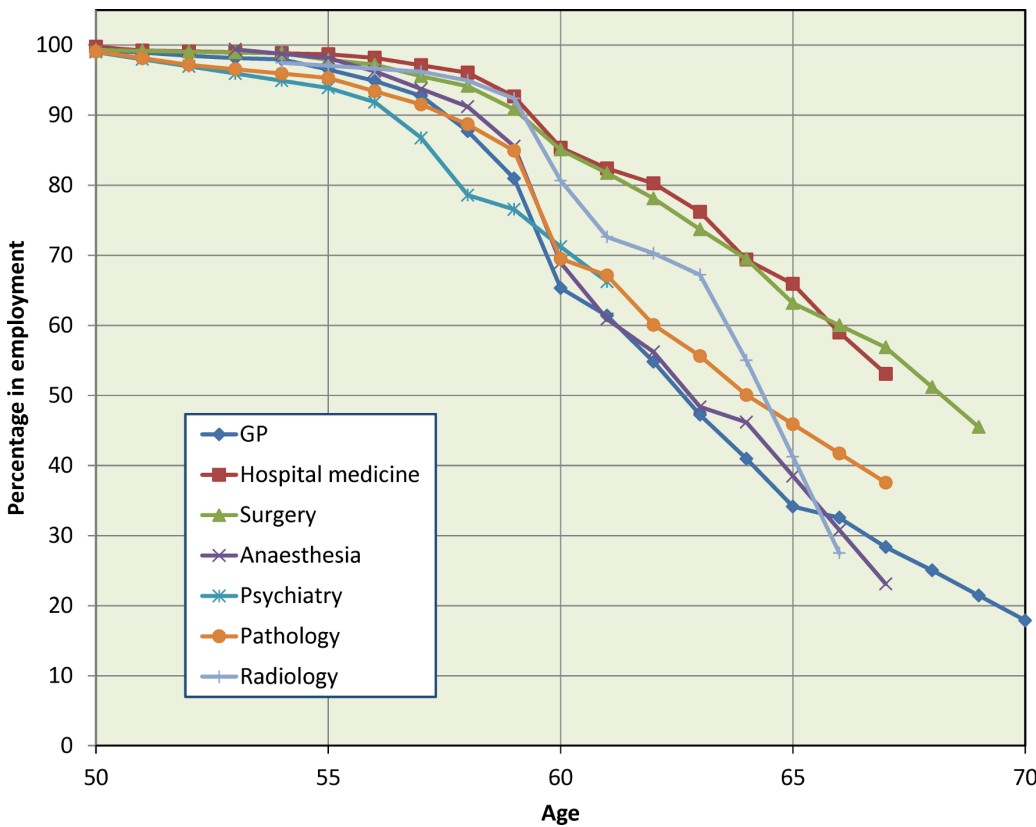

**Figure 2** Male 1974 and 1977 graduates still working in medicine, percentages by age: Kaplan-Meier estimates.

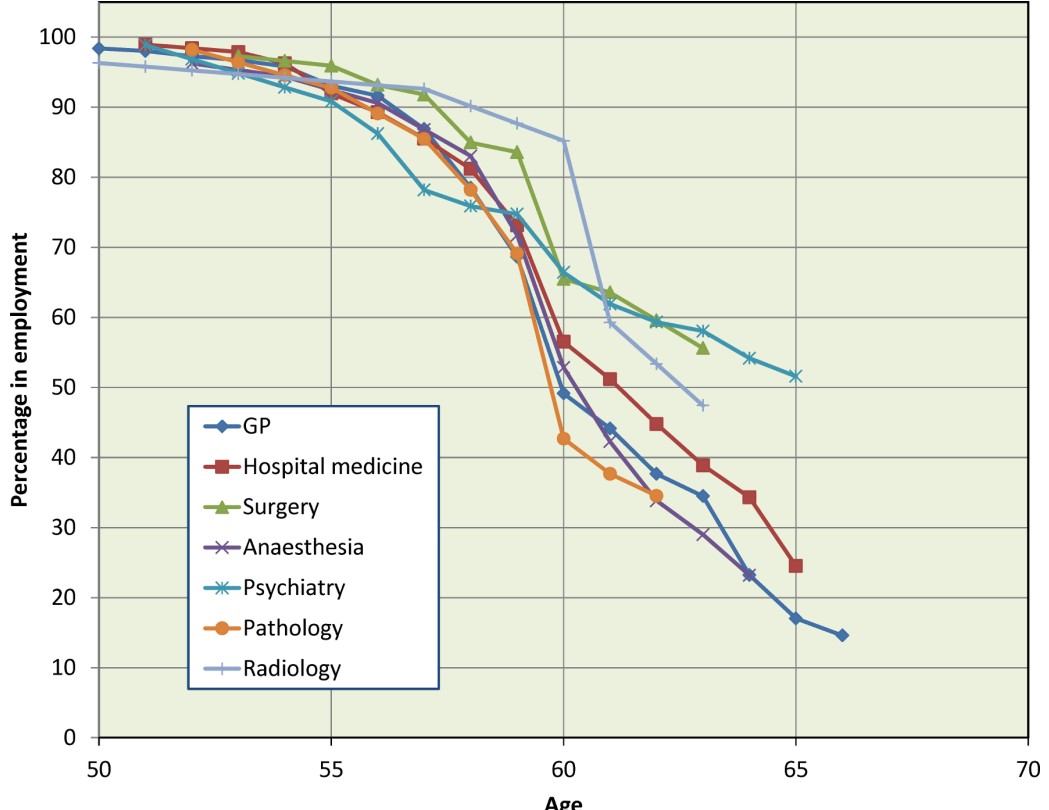

**Figure 3** Female 1974 and 1977 graduates still working in medicine, percentages by age: Kaplan-Meier estimates.

the same extent to younger generations when they reach the equivalent stages in their careers.

Retirement profiles by age showed similar results and, additionally, in general practice, the hospital medical specialties, surgery, anaesthesia and pathology, the proportion of women who had retired increased more with increasing age than was the case for men.

Among those who had retired early, changes in the work environment and in personal circumstances were often contributory factors.

GPs who were still working were more inclined to have planned work reductions than their hospital colleagues.

### Strengths and limitations

This is a large-scale study of graduates from all UK medical schools. The study has a very high response rate (85%) for a questionnaire study with voluntary participation. However, there is a possibility of non-responder bias.

In 2017, in the UK, 47% of the registered UK medical workforce were women: of doctors aged under 30 years (13% of the whole workforce), 62% were women.[16] By contrast, in our study, women constituted only 30% of the 1974/1977 cohorts. We cannot judge whether younger cohorts, with higher percentages of women, will eventually show similar profiles of retirement to the subjects in our study.

### Comparison with existing literature

The doctors we surveyed retired, on average, at age 60—2 years earlier than the average intended retirement age given by consultants in a UK census conducted in 2014–2015.[5] The UK census, however, comprises consultants who are older than the 1974 and 1977 cohorts we studied. On 1 February 2018, Pulse Magazine reported that analysis of figures received for NHS Business Services Authority under a Freedom of Information request showed that 62% of GPs claiming their pension for the first time in 2016/2017 were under 60 years, compared with 33% in 2011/2012. A review of studies which reported both intended and actual retirement ages found the two ages to be similar.[17] As mentioned in the Introduction section, there is a general societal trend towards later retirement (which contrasts with our finding of doctors' earlier retirement), with UK women in particular working longer as the state pension age increases.[18–20] An earlier analysis of these cohorts, reporting a median age of 64 for those who retired in 2014, found that 27% had retired earlier than they had originally intended and, of those who had retired, their main reasons for retirement were 'increased time for leisure/other interests' and 'pressure of work'.[21] A systematic review of physician retirement identified burnout and excessive workload as the main reasons for early retirement.[17]

We found that more GPs (54%) than doctors in other specialties were retired, and 63% of women GPs were retired. Factors influencing GPs' intentions to leave general practice include workload intensity, bureaucracy, appraisal and revalidation.[22–24] Research in Australia of doctors aged over 55 found that psychiatrists and GPs

were less likely than doctors in other specialties to say that they intended to retire,[25] and research in Canada found no difference in retirement age between specialties.[26] In the UK, GPs are reported to have had higher levels of burnout than doctors in other specialties.[27]

## Implications/conclusions

Our findings demonstrate the scale of loss of doctors from the workforce as a result of retirement in the early 60s. Premature loss of experienced doctors in late career has an adverse impact on workforce capacity and it needs to be better understood. Nonetheless, it is difficult to untangle the motivations of doctors as they consider and plan the reality and timing of their retirement. Every retirement, like every medical career, will be individual and will be the outcome of a number of considerations related to work, home life and leisure. In these cohorts, women are retiring younger than men and there was substantial variation by specialty. There is scope for further study of underlying reasons for early retirement, and in particular to address the question of whether those attracted to some specialties are more likely than others to have characteristics which predispose them to early retirement, or whether there are inherent issues in some specialties which render early retirement more attractive. If the latter, policy amendments to ameliorate these issues may reduce workforce losses consequent to early retirement.

**Acknowledgements** We thank Shelly Lachish for early work on the paper and Janet Justice and Alison Stockford for data entry. We are very grateful to all the doctors who participated in the surveys.

**Contributors** FS performed the analysis and wrote the first drafts of the paper. MJG and TWL designed and conducted the surveys. All authors contributed to further drafts and all approved the final version.

**Funding** The report is based on independent research commissioned and funded by the NIHR Policy Research Programme (project number 016/0118).

**Disclaimer** The views expressed in the publication are those of the author(s) and not necessarily those of the NHS, the NIHR, the Department of Health, its arm's length bodies or other government departments.

**Competing interests** None declared.

**Patient consent** Not required.

**Ethics approval** National Research Ethics Service, following referral to the Brighton and Mid-Sussex Research Ethics Committee in its role as a multicentre research ethics committee (ref 04/Q1907/48 amendment Am02 March 2015).

**Provenance and peer review** Not commissioned; externally peer reviewed.

**Data sharing statement** It may be possible for the authors to make tabulated data, produced in the course of this work but not included in the paper, available to interested readers on request.

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
