## [Reviewer comments · BMJ Open]

ARTICLE DETAILS

TITLE (PROVISIONAL)	Retirement ages of senior UK doctors: national surveys of the medical graduates of 1974 and 1977
AUTHORS	Smith, Fay; Goldacre, Michael; Lambert, Trevor

VERSION 1 – REVIEW

REVIEWER	Dr Sharon Spooner University of Manchester, UK
REVIEW RETURNED	20-Mar-2018

GENERAL COMMENTS	This is an important area wrt current and projected patterns of work in the NHS and highlights some significant issues for further investigation. However, I believe that the term 'contractual retirement age' is ill-defined in the paper and warrants clarification or revision. This term seems to imply that a specific or fixed retirement age is stated in the employment contracts of all responding NHS consultants and GPs. However, current NHS consultants tell me that they are unaware of any contractual retirement age in their contracts; their retirement expectations are more closely aligned with pension entitlements than their employment contracts. Similarly, NHS GP partners do not generally have an employment contract; their Partnership Agreements are generally constructed to fit the requirements of their Practice circumstances and preferences and may or may not make mention of an expectation about the age of retirement from the partnership (from experience, this has generally been in the form of a clause which stated 'must retire from the partnership by the age of 70 years unless otherwise agreed'). It may be that older consultant contracts differ from current contracts in this respect, but I feel that clarification of what is meant by 'contractual retirement age' (e.g as opposed to 'intended retirement age') and how it varies across specialties would be essential in the revision of this paper. Further, the text does not clearly indicate how the authors were able to be certain that doctors who did not respond to the invitation to participate had received the invitation (i.e. were contacted).
--

REVIEWER	Bryan McIntosh University of Bradford, UK
REVIEW RETURNED	31-Mar-2018

GENERAL COMMENTS	Very competent and strong paper. Timely and I have no hesitation in accepting.
--

VERSION 1 – AUTHOR RESPONSE

Reviewers' Reports:

Reviewer: 1

Reviewer Name: Dr Sharon Spooner

Institution and Country: University of Manchester, UK

Competing Interests: None declared

This is an important area wrt current and projected patterns of work in the NHS and highlights some significant issues for further investigation.

However, I believe that the term 'contractual retirement age' is ill-defined in the paper and warrants clarification or revision. This term seems to imply that a specific or fixed retirement age is stated in the employment contracts of all responding NHS consultants and GPs. However, current NHS consultants tell me that they are unaware of any contractual retirement age in their contracts; their retirement expectations are more closely aligned with pension entitlements than their employment contracts. Similarly, NHS GP partners do not generally have an employment contract; their Partnership Agreements are generally constructed to fit the requirements of their Practice circumstances and preferences and may or may not make mention of an expectation about the age of retirement from the partnership (from experience, this has generally been in the form of a clause which stated 'must retire from the partnership by the age of 70 years unless otherwise agreed'). It may be that older consultant contracts differ from current contracts in this respect, but I feel that clarification of what is meant by 'contractual retirement age' (e.g as opposed to 'intended retirement age') and how it varies across specialties would be essential in the revision of this paper.

RESPONSE: ON REFLECTION, WE AGREE WITH THE REVIEWER'S RESERVATIONS ABOUT THE NOTION OF A CONTRACTUAL RETIREMENT AGE. THIS HAS WE BELIEVE CHANGED OVER THE YEARS AND THESE COHORTS STRADDLE THE CHANGE AWKWARDLY – EARLIER RETIRERS MAY HAVE RETIRED UNDER A DIFFERENT CONTRACTUAL SITUATION FROM LATER RETIREES. FOR THOSE STILL WORKING IN MEDICINE, WE DID ASK (THOUGH WE DID NOT GIVE DETAILS IN THE DRAFT) WHETHER THE DOCTOR'S CURRENT POST HAD A FIXED (EMPLOYER JUSTIFIED) RETIREMENT AGE – AND THE RESULTS TO THAT PRELIMINARY QUESTION CONFIRM THAT FOR MANY CONTINUING DOCTORS THEIR CONTRACTS DID NOT SPECIFY A FIXED RETIREMENT AGE.

WE HAVE THEREFORE DECIDED TO AMEND THE PAPER TO REMOVE MENTION OF CONTRACTUAL RETIREMENT AGES ENTIRELY, AND TO FOCUS EXCLUSIVELY ON ACTUAL RETIREMENT AGES. WE DON'T THINK THE PAPER SUFFERS OR LOSES ANYTHING IMPORTANT AS A RESULT.

Further, the text does not clearly indicate how the authors were able to be certain that doctors who did not respond to the invitation to participate had received the invitation (i.e. were contacted).

RESPONSE: CONTACT WAS BY EMAIL AND POST, ON SEVERAL OCCASIONS EACH, USING CURRENT CONTACT DETAILS PROVIDED BY THE GMC – WE HAVE ADDED A PHRASE IN THE FIRST PARAGRAPH OF METHODS TO CONFIRM THIS. WE BELIEVE THE GMC DATA OFFER A HIGH DEGREE OF ACCURACY. FAILURES TO CONTACT WILL BE LIKELY TO BE RELATED TO DOCTORS WHOSE GMC REGISTRATION HAD LAPSED. WE ACKNOWLEDGE IN STRENGTHS AND LIMITATIONS THAT THIS IS A POSSIBILITY.

Reviewer: 2
Reviewer Name: Bryan McIntosh
Institution and Country: University of Bradford, UK
Competing Interests: None declared

Very competent and strong paper. Timely and I have no hesitation in accepting.

RESPONSE: WE THANK THE REVIEWER FOR HIS KIND COMMENT.

FORMATTING AMENDMENTS (if any)

Required amendments will be listed here; please include these changes in your revised version:
- Kindly re-upload SUPPLEMENTARY TABLE in PDF format.

RESPONSE: WE WILL DO THIS.

- Patient and Public Involvement statement:

Authors must include a statement in the methods section of the manuscript under the sub-heading 'Patient and Public Involvement'.

This should provide a brief response to the following questions:

How was the development of the research question and outcome measures informed by patients' priorities, experience, and preferences?

How did you involve patients in the design of this study?

Were patients involved in the recruitment to and conduct of the study?

How will the results be disseminated to study participants?

For randomised controlled trials, was the burden of the intervention assessed by patients themselves?

Patient advisers should also be thanked in the contributorship statement/acknowledgements.

If patients were not involved please state this.

RESPONSE: WE HAVE INCLUDED THE FOLLOWING AFTER ACKNOWLEDGEMENTS:

PATIENT AND PUBLIC INVOLVEMENT: PATIENTS WERE NOT INVOLVED IN THE DESIGN OR ANY ASPECT OF THE STUDY, BY AGREEMENT WITH THE FUNDING BODY, SINCE THE STUDY DID NOT INVOLVE ANY MEDICAL OR PATIENT DATA. RESULTS OF THE STUDY ARE PUBLISHED IN VARIOUS PAPERS IN THE PEER-REVIEWED LITERATURE, AND SUMMARY REPORTS OF OUR SURVEY WORK ARE ON OUR WEBSITE AT WWW.NDPH.OX.AC.UK

VERSION 2 – REVIEW

REVIEWER	Dr Sharon Spooner University of Manchester
REVIEW RETURNED	24-May-2018
GENERAL COMMENTS	This revisions in this paper resolve the main issue highlighted in my review of a previous manuscript.